# Mechanisms of Activation of Brain’s Drainage during Sleep: The Nightlife of Astrocytes

**DOI:** 10.3390/cells12222667

**Published:** 2023-11-20

**Authors:** Dmitry Postnov, Oxana Semyachkina-Glushkovskaya, Elena Litvinenko, Jürgen Kurths, Thomas Penzel

**Affiliations:** 1Department of Optics and Biophotonics, Saratov State University, Astrakhanskaya Str. 83, 410012 Saratov, Russia; ells03@yandex.ru; 2Department of Biology, Saratov State University, Astrakhanskaya Str. 83, 410012 Saratov, Russia; glushkovskaya@mail.ru (O.S.-G.); juergen.kurths@pik-potsdam.de (J.K.); 3Physics Department, Humboldt University, Newtonstrasse 15, 12489 Berlin, Germany; 4Potsdam Institute for Climate Impact Research, Telegrafenberg A31, 14473 Potsdam, Germany; 5Charité — Universitätsmedizin Berlin, Charitéplatz 1, 10117 Berlin, Germany

**Keywords:** astrocytes, brain waste removal system, neuro-glial-vascular unit, local sleep, noradrenaline

## Abstract

The study of functions, mechanisms of generation, and pathways of movement of cerebral fluids has a long history, but the last decade has been especially productive. The proposed glymphatic hypothesis, which suggests a mechanism of the brain waste removal system (BWRS), caused an active discussion on both the criticism of some of the perspectives and our intensive study of new experimental facts. It was especially found that the intensity of the metabolite clearance changes significantly during the transition between sleep and wakefulness. Interestingly, at the cellular level, a number of aspects of this problem have been focused on, such as astrocytes–glial cells, which, over the past two decades, have been recognized as equal partners of neurons and perform many important functions. In particular, an important role was assigned to astrocytes within the framework of the glymphatic hypothesis. In this review, we return to the “astrocytocentric” view of the BWRS function and the explanation of its activation during sleep from the viewpoint of new findings over the last decade. Our main conclusion is that the BWRS’s action may be analyzed both at the systemic (whole-brain) and at the local (cellular) level. The local level means here that the neuro-glial-vascular unit can also be regarded as the smallest functional unit of sleep, and therefore, the smallest functional unit of the BWRS.

## 1. Introduction

Sleep occurs in every animal that has a nervous system, including humans, birds, fish, flies, and even worms [1]. “According to a simple behavioral definition, sleep is a reversible behavioral state of perceptual disengagement from and unresponsiveness to the environment” [2]. Humans spend about one-third of their lives sleeping. If we do not get enough sleep, the effects of a number of processes increase, e.g., inflammation, accumulation of protein waste, excitotoxicity, etc., which rapidly deteriorate the brain. Indeed, chronic sleep loss is accompanied by astrocytic phagocytosis of synaptic elements leading to microglia activation [3,4]. Even one night without sleep leads to the accumulation of amyloid-beta (Aβ) in the brain tissue of healthy people [5,6]. Obviously, sleep is necessary for the health of the central nervous system protecting against the development of various diseases, including brain pathology [7]. Therefore, insufficient sleep (6 h or less per night) over 25 years was accompanied by the development of dementia in a large group of 8000 volunteers aged 50–60 years [8].

However, the mechanisms underlying the restorative function of sleep remain unknown. A widespread belief is that the main function of sleep is to recharge our energy. Many theories explaining the restorative properties of sleep have been proposed [1,9,10]. Since the pioneering work [11], a number of studies have contributed to the understanding of the biological role of sleep in removing metabolites and toxins from the central nervous system through the brain waste removal system (BWRS) [12,13,14,15,16,17,18]. The relationship between BWRS and sleep was first described in [11], where the brain influx of CSF tracers in mice during sleep, awake, and under anesthesia was measured [11].

Alzheimer’s disease (AD) is accompanied by both poor sleep and an increased Aβ deposition in the brain [19]. Ref. [11] clearly demonstrated a 95% reduction in the Aβ removal from the brain during wakefulness and the activation of this process during deep sleep due to a 60% increase in the extracellular space (ECS) in parenchyma. It is important to note that the Aβ content in the cerebral spinal fluid (CSF) is higher at night before sleep and lower in the morning after sleep [5,11]. Currently, sleep is considered a biomarker and a promising therapeutic target for AD [20,21,22,23,24,25,26]. There is common agreement that sleep accelerates a removal of Aβ from the brain [5,11], while the underlying physiological mechanisms are still under study. In [27], it was confirmed that sleep causes an increase in the BWRS activity arising from the expansion of the ECS. In [12], it was reported that deep sleep is accompanied by high oscillations of CSF in the human brain. In [28], it was found that both CSF distribution over the brain and the expressions of astrocytic aquaporins AQP4 are under circadian control. It is being actively discussed whether the nighttime activation of the BWRS is a main driving factor for the removal of metabolic wastes from the sleeping brain [12,13,14,15,16,17,18,27].

In our experiments (data not published), using multiphoton microscopy for real-time monitoring of the BWRS in non-anesthetized mice under EEG control with further ex vivo confocal imaging of the whole brain (Figure 1a), we found a significant increase in the perivascular spaces (PVSs) during deep sleep (Figure 1b–e). To analyze the activation of the BWRS during deep sleep, 2% fluorescein isothiocyanate-dextran 70 kDa (FITCD, green) was automatically injected into the right lateral ventricle via a chronically implanted polyethylene catheter in the time of monitoring of delta band EEG activity. The cerebral vessels were filled with 1% Evans Blue dye (EBD, red), which was injected via a polyethylene catheter implanted into the femoral vein. Figure 1d,e (ex vivo data) clearly demonstrates that the intensity of the fluorescent signal from FITCD was higher in the sleeping vs. waking brain. The brighter signal from FITCD during sleep can be explained by the expansion of the space around the cerebral vessels (in vivo data, Figure 1b,c). Figure 1f illustrates the hypothesis that during deep sleep the size of PVSs and the volume of interstitial fluids increase, promoting the removal of metabolites from brain tissues. During wakefulness, the size of PVSs decreases and the exchange between fluids and brain tissues is suppressed.

The figure and discussion above strongly suggest that sleep-related changes in the clearance of harmful metabolites are clearly expressed at the level of small cellular structures. However, the specific cellular mechanisms and signaling pathways underlying the sleep-mediated changes in the BWRS activity and brain fluid movement remain poorly understood.

Astrocytes regulate many important mechanisms of neural homeostasis. Recently, the role of astrocytes in sleep regulation has become apparent due to significant advances in brain imaging, the use of transgenic mice, and optogenetics. These new data show that astrocytes change their activity across the sleep–wake cycle and may control sleep need via changes in intracellular signaling pathways [29]. Astrocytes also modulate sleep architecture through the secretion of sleep-inducing molecules [29]. Indeed, several in vivo studies clearly demonstrate that sleep quality is modulated by the astroglial transport of signaling molecules [30,31,32]. The astroglial intracellular calcium-dependent signaling pathways, as well as cyclic adenosine monophosphate, regulate sleep time [33,34,35,36]. Both ex vivo and in vivo studies revealed that the astroglial calcium activity encodes sleep need [33,37].

In this state-of-the-art review, we discuss these findings in more detail as well as further interactions between astrocytes and sleep. In doing so, we follow an “astrocytocentric” view of BWRS research, with an emphasis on the mechanisms associated with its activation during sleep. Thus, in Section 2, we discuss experimental data and established mechanisms of the influence of astrocytes on sleep. In Section 3, we highlight the concept of a neurovascular unit, which recently was suggested to be extended to an “active milieu”, which includes in a functional unit not only the neuron, astrocyte, and blood vessel, but also tripartite synapses and intercellular spaces. There is evidence that such an active milieu can be regarded as a functional unit of the sleep–wake transition that, in turn, affects the clearance of harmful metabolites. In Section 4, we discuss mechanisms of astrocyte volume regulation. In particular, we highlight that swelling and contraction of astrocytes within the neurogliavascular unit drive BWRS activity. In Section 5, we discuss control signals and mechanisms that provide a change in the astrocyte volume during the transition between sleep and wakefulness. In Section 6, we describe the physical aspects of BWRS functions and various mechanisms that can lead to astrocyte shrinking during sleep. In Section 7, we discuss the limitations of research on the review topic related to the possible contribution of sexual dimorphism. Finally, in Section 8 (Conclusions), we present a condensed formulation of the outcome of our consideration.

## 2. Mechanisms of Modulation of Sleep by Astrocytes

Astrocytes are found in various areas of the brain and therefore can affect sleep globally, through arousal centers, and locally, at the cell-to-cell communication level [31,38,39,40]. It has been repeatedly shown that the activity of astrocytes is significantly different in the states of sleep and wakefulness. Although all these changes should be regarded as different features of one complex system, below we group the results according to the signaling molecules that were measured.

### 2.1. Astroglial Calcium

Calcium in the astrocyte cytosol plays the role of a universal messenger, responding to various stimuli and triggering a variety of response mechanisms [41,42,43,44]. The review by [33] collected evidence that astrocytic calcium changes dynamically with sleep, wake, and sleep loss, and encodes changes in sleep needs. It was also shown that the synchrony of calcium events in the astrocyte network decreases during sleep compared to wakefulness. It was noted that the calcium activity of astrocytes is reduced during sleep compared to wakefulness, and its increase precedes transitions from sleep to wakefulness [35]. The authors of [45] emphasize the difference in spatiotemporal patterns of astrocytic calcium activity in the murine barrel cortex. While sleep is characterized by low-intensity prolonged elevations of calcium, widespread short-lasting calcium spikes are typical for wakefulness. Large-scale calcium waves in aroused mice are inositol triphosphate (IP3)-dependent, evoked mostly by the sensory input, and contributing to reliable sensory transmission. Localized calcium spikes appeared to be IP3-independent and associated with decreased extracellular potassium, hyperpolarization of the neurons, and suppression of sensory transmission.

It has now been established that astrocyte dysfunction, in general, and disruption in their calcium dynamics, in particular, are reliably associated with AD [46,47,48,49,50] or cognitive impairment [51,52].

Calcium can, therefore, be regarded as a measure of the level of astrocyte activity and, to a large extent, as an indicator of the current state (sleep or wakefulness) of the entire organism—both in health and in disease.

### 2.2. Adenosine-Mediated Pathway

Currently, there is a consensus that astrocytes are actively involved in the regulation of the sleep–wake cycle through gliotransmission. The most studied and confirmed mechanism is based on adenosine as a neurotransmitter.

The work [53] showed that astrocytes can release ATP both in tonic mode, which leads to a permanent suppression of synaptic transmission, and in the form of phasic release, which modulates synaptic plasticity when activity-dependent recruitment of astrocytes occurs. It was demonstrated that glia-derived adenosine is responsible for activity-dependent heterosynaptic depression at excitatory synapses. In [31], the authors genetically inhibited the release of gliotransmitters to examine whether astrocytes play a significant role in sleep regulation. Inhibition of gliotransmission was shown to attenuate sleep pressure accumulation, as assessed by measuring EEG slow wave activity during non-rapid eye movement (NREM) sleep, and prevent cognitive impairment associated with sleep loss. It was also shown that the impairment of vesicular gliotransmission attenuates cortical slow oscillations [54]. The reduction in slow oscillations was confirmed by per-moment EEG recordings in freely behaving mice during natural sleep [32].

In the review [55], the authors discuss the contribution of astrocytes in the context of conceptual models of sleep generation and functioning, which have historically focused on neural mechanisms. They conclude that there are two different aspects of gliotransmission that must be taken into account when building hypotheses and models, namely temporal and spatial complexity of astrocytic neuromodulatory feedback to networks. In [56], dnSNARE mice were used to check whether astrocytes contribute to the increased sleep pressure during immune loading and whether this is the result of changes in adenosine signaling. It has been shown that dnSNARE-mediated gliotransmission is required for the ability of lipopolysaccharide to increase sleep pressure, as measured by the power of slow-wave activity during NREM sleep. The study [57] demonstrated that the antidepressant effect of sleep deprivation depends on the ability of astrocytes to regulate extracellular adenosine, thereby affecting sleep. The paper [58] discussed various roles of astrocytes during sleep, including modulation of the sleep homeostasis process through the release of adenosine, which acts on adenosine receptors A1 and promotes sleep. In [59,60], a specific mechanism of action of adenosine was proposed, including the interaction of astrocytes and neurons in the regulation of sleep, in which endogenous adenosine derived from astrocytes excites sleep-promoting neurons and thus reduces the excitability of neurons in brain regions associated with awakening.

Taken together, it has been shown that astrocytic adenosine, acting through A1 receptors, contributes to the modulation of sleep pressure. The relative roles of these processes in sleep homeostasis are essentially unknown.

### 2.3. Non-Adenosine Pathways

In [61], it was reported that direct stimulation of astrocytes powerfully induces sleep during the active phase of the sleep–wake cycle. It is hypothesized that optogenetic stimulation of astrocytes released cytokines, which then affected the orexin and MCH neurons regulating sleep. In [1], it has been indicated that in addition to the adenosine pathway, astrocytes and oligodendrocytes can have an effect mediated by voltage-dependent external potassium currents. The authors of [34] examined the astrocytic network, comprising a cortex-wide syncytium that influences the population-level neuronal activity. They find that different astrocytic G-protein-coupled receptor (GPCR) signaling pathways separately control the NREM sleep depth and duration and that astrocytic signaling causes differential changes in the local and remote cortex. These data support a model in which the cortical astrocyte network serves as a hub for regulating distinct NREM sleep features. Some evidence suggesting that modulation of sleep–wake behavior by astrocytes can occur without adenosine signaling can be found in [62]. According to [63], astrocytes detect neuronal signals released during wakefulness, integrate those signals via changes in intracellular calcium, and, via negative feedback, dampen those waking signals, resulting in an increased slow-wave activity during NREM and sleep time.

Summarizing the above, astrocytes have the ability to communicate with brain structures that control sleep. Changes in astrocyte calcium activity are closely associated with sleep needs—the canonical measure of which is considered to be slow wave activity (SWA) [31,64]. This connection depends on the brain region, but in general, can be found in multiple locations [33,55,62].

The above is quite consistent with the research on “local sleep”, which we discuss in the next section and allows us to hypothesize that a local increase in sleep need may signal excessively high neural activity. Indeed, the mechanisms of the astrocyte calcium response to the synaptic activity of a neuron are well known, for example, [65,66]. The concept of gliotransmitter signaling, including the discussed adenosine pathway, suggests that this signal is particularly strong at high levels of neural activity, which, in turn, is associated with the local depletion of metabolic resources. Therefore, it is reasonable to hypothesize that such a signaling pathway may be directed toward increasing the need for sleep under conditions of local neuronal ”overload”.

### 2.4. Astrocytes Participate in Circadian Timekeeping

Currently, there is increasing evidence that the suprachiasmatic nucleus (SCN) circuit includes astrocytes as essential time-keepers.

The authors of [67] discuss the involvement of astrocytes in the regulation of the circadian rhythm. It is argued that SCN circuit-level timekeeping arises from interdependent and mutually supportive astrocytic-neuronal signaling since the somatic genetic re-programming of intracellular clocks in the SCN astrocytes was capable of remodeling circadian behavioral rhythms in adult mice. The review article [68] presents the current view of the SCN circuit and discusses whether astrocytic functions described in other brain regions could help explain those well- and not-so-well-known features of the central pacemaker. A recent review article by Hastings et al. [69] takes the next step, showing the connection between rhythm disturbances and the development of pathologies. Thus, in mouse models of AD, circadian disturbances accelerate astroglial activation and other brain pathologies. In brain cancer, treatment in the morning has been associated with prolonged survival, suggesting that circadian time is fast becoming critical to elucidating reciprocal astrocytic-neuronal interactions in health and disease.

## 3. Neuro-Glia-Vascular Unit Is also Local-Sleep Unit and Local Brain-Drainage Unit

As noted above, the sleep–wake cycle is a two-way street. In such feedback systems, it can be difficult to isolate cause and effect, but it is always useful to break the system down into separate functional blocks. Applied to cortical astrocytes and neurons, this approach was formalized two decades ago in the form of the “neurovascular unit”.

### 3.1. Neuro-Glia-Vascular Unit

The concept of a neurovascular unit took shape in 2001 when it became clear that meeting the metabolic needs of a neuron is actively regulated at the local level. This concept refers to a structure in which a single astrocyte handles the request of a neuron, initiating a vasodilating response when it is highly active, and also provides delivery of glucose to the neuron, while oxygen reaches the neuron through diffusion [70,71,72]. A number of modeling studies have been devoted to simulating the main functions of the neurovascular unit [73,74,75,76].

With the accumulation of data, it became clear that both the functions of the neurovascular unit and the role of the astrocyte in it are much more diverse than it was initially thought. In particular, the concept of a tripartite synapse [77,78] has been developed, according to which thin processes of astrocytes, tightly covering the synapse, intercept leaking glutamate and thus monitor the level of the current activity of neuronal activity and respond to it both dynamically, generating calcium bursts, and long-term, providing neuronal plasticity [79,80,81,82,83].

By now, it has been established that the neurovascular unit is also sensitive to quite moderate physiologically normal changes in neuronal activity, such as the transition between sleep and wakefulness [29,58,84,85].

Since extrasynaptic transmission is important in all these processes, it is logical to consider a fragment of the intercellular space associated with the neurovascular unit as its part. The growing understanding of the complexity and diversity of connections in the neurovascular unit logically led to its expansion to the neuro-glia-vascular unit (NGVU), which now includes the neuron, astrocyte, pericytes, and endothelial cells of the blood vessel, as well as the extracellular space and extracellular matrix [86,87]. In [87], Semyanov and Verkhratsky proposed a new term, “active milieu”, and the above set included neuronal and glial compartments, extracellular space, extracellular matrix, and vasculature.

For the topic of our review, it is extremely important that the NGVU has a clear spatial localization [88]. According to [89,90], an astrocyte has the shape of a cloud with many thin processes, and the areas occupied by neighboring astrocytes do not overlap much.

In addition, this astrocyte morphology provides a very high ratio of cell membrane area to extracellular space (ECS) volume, which, for the parenchyma, is in the range of 10–30% of the total tissue volume.

Thus, the NGVU can be regarded as a spatially isolated structure that receives external signals (e.g., noradrenaline) and, in response, changes its state according to the current level of neuronal activity. We will return to a discussion of these ways of responding below, and in this section, we will discuss what the important implications of this isolation are.

### 3.2. NGVU as Sleep Unit

For a long time, the state of sleep was considered a global state of the brain, needed only by the brain itself and controlled exclusively by neural connections between neuron nuclei, forming a “sleep switch” [91,92]. However, the emergence of data on unihemispheric sleep in aquatic mammals and the accumulation of knowledge about the physiology of the brain led to the concept of local sleep, which, at the moment, can be considered quite reasonable [93,94,95]. According to this concept, not all areas of the brain may be in a physiological state corresponding to sleep. In [94], the authors wrote: “…taking into account the neuronal firing pattern, sleep onset (SO) may occur in a strictly local manner; the animal can be behaviorally awake, but local cortical “islands of sleep” can appear”. It should be especially noted that we are talking about both populations and individual neurons. It has been shown that in sleep-deprived monkeys performing a visual discrimination task, neurons in the striate cortex showed a waking pattern of activation, but some neurons in the extrastriate visual cortex displayed a characteristic sleep firing pattern [96]. Thus, there are no obstacles to considering the concept of local sleep at the level of one single NGVU.

Here, the results of [84,97] provide encouraging help, according to which the ionic composition of interstitial fluid changes significantly during sleep, as expected. More importantly, when an artificial “sleepy” intercellular fluid is applied, the brain tissues locally go into a state physiologically similar to sleep [84].

Since the local sleep is detected and measured by means of SWA, the results described above refer to neural activity. However, as mentioned above, the degree of manifestation of SWA depends on the activity of astrocytes [29,33]; therefore, their participation in local sleep is implied.

The results described above explain that sleep can be interpreted as a specific state of cells that form the neurogliavascular unit, and the NGVU itself can be regarded as the smallest “sleep unit”, at the level of which the transition between sleep and wakefulness occurs.

### 3.3. Why NGVU Is Also a Drainage Unit

Based on the above, it is logical to interpret NGVU as the minimum possible cellular structure of BWRS. Indeed, “individual astrocytes tend to form nonoverlapping domains, placing them at the center of regulating local homeostatic functions” [88]. An integral part of each NGVU is the access to the blood vessel [86,87], and hence to the perivascular space, which, in turn, serves as the BWRS conduit.

Fellin et al. wrote: “Astrocytes are characterized by a highly ramified structure of cellular processes that occupy non-overlapping domains” [55]. The volume occupied by the astrocyte is connected to the perivascular space through gaps between the endfeet of the astrocyte. Astrocytes are networked and support the spread of calcium activity from cell to cell. When studying astrocyte cultures, various phases of wave activity were discovered, with the number of cells covered by a single wave before its disappearance ranging from 3 to sometimes 8–10 [98]. According to the review work of [99], the number of cells captured in one wave ranged from 100 to 400 cells, depending on the region of the brain where the astrocytes were taken from and the method of stimulating them. There is, however, evidence that calcium waves observed in vivo are rarely spread more than 80 μm [100,101]. Thus, intercellular calcium waves cover an area that is small in terms of activity patterns measured by EEG and can be considered a local phenomenon.

These data support the assumption that NGVU (or a small ensemble of them) can be seen as a self-contained “drainage unit”, subordinate to the whole-brain mechanisms, which, in turn, supply fresh cerebrospinal fluid and create arterial and other pulsations [12].

From the point of view of physical laws, facilitating drainage during sleep is a consequence of a significant increase in the proportion of intercellular space. Therefore, the way to understand BWRS activation lies in understanding how the proportion of ECS in NGVU is regulated in general and during the transition between sleep and wakefulness, in particular.

## 4. Astrocyte Volume Regulation

There are many experimental data on astrocyte swelling or shrinking under the influence of osmotic forces in various situations [102,103,104,105,106]. At the same time, it is noted that neurons swell to a lesser extent and restore their volume faster [107,108,109,110]. The recent review article [111] provides a good overview of the mechanisms of astrocyte volume regulation.

Early hypotheses were based on potassium buffering, then siphoning, and relied on inward-rectifier potassium channels and on AQP4 as a channel for passing water into the astrocyte [112,113,114,115].

To date, it is clear that the situation is more complicated, and at different levels of extracellular potassium rise, various mechanisms are activated, (see Figure 5 in [111]). Specifically, when there is a local and transient increase in extracellular potassium levels due to local synaptic activity, it can enter astrocyte processes through KIR channels, while water moves through AQP4. When potassium levels increase to 5–6 mM, monocarboxylate transporters, NBCe1 and Na${}^{+}$/K${}^{+}$ ATPase, may contribute to the increase in the intracellular potassium concentration in conjunction with the marked increase in the astrocyte volume under these conditions. At very high levels of extracellular potassium, such as in seizure or even higher, a strong cellular depolarization activates the sodium–potassium–chloride cotransporter NKCC1 and reverses potassium–chloride cotransporter KCC, resulting in the rapid ion influx and associated severe cell swelling.

In relation to sleep and awake states, we are interested in the range between normally low (sleep) and normally high (wakefulness) levels of extracellular potassium.

As can be expected, astrocyte contraction during the transition to sleep is in the region of regulation by the sodium–potassium pump (Na${}^{+}$/K${}^{+}$ ATPase, NKA). Recent studies using MRI indicate that there is considerable water movement tied to NKA activity [116,117].

However, this is not entirely clear, since the operation of the pump itself creates an outward osmotic gradient. It is most likely that the change in the astrocyte volume is not due to the transfer of ions by the pump itself but to multiple processes associated with changes in NKA activity: “… must be other processes working hand-in-hand with the NKA to produce a net increase in intracellular osmolarity in conditions of elevated potassium” [111].

The decrease in intracellular sodium corresponded to only about 10% of the accumulated intracellular potassium [118]. This is likely because the inward movement of sodium ions through other membrane transport processes is tightly coupled with pump activity; otherwise, the pump would rapidly become ineffective at potassium removal as intracellular sodium falls [119,120]. Some influx or reduced efflux of chloride due to the potassium-induced membrane depolarization, or production of endogenous anionic metabolites, such as bicarbonate and lactate [118,121], are also regarded as relevant mechanisms.

It is reasonable to consider the contraction of astrocytes during sleep as an inversion or simply a cessation of those processes that provide swelling with moderate activation of neurons. Then, by itself, the work of the NKA can provide astrocyte shrinking.

It has been repeatedly noted that changes in the astroglial chloride concentration are associated with changes in the astrocyte cell volume associated with activity [122,123,124]. The review paper [125] summarizes that one of the important functions of astroglial chloride is the regulation of cell volume and, hence, morphological plasticity. Currently a number of swelling-activated anion channels in glial cells and neurons are known [126,127,128]. When removing a large amount of excessive extracellular potassium, astrocytes take up large amounts of anions, mostly chloride [129,130]. The clearance of excessive extracellular potassium due to high neuronal activity by astrocytes leads to osmotic gradients, resulting in water influx through aquaporins and astrocytic dilation [131,132,133]. According to this mechanism, the high activity of the neuron, in which the astrocyte ensures the removal of excess potassium, is accompanied by an inward flow of chloride ions into the astrocyte and dramatically increases astrocyte swelling. This path was implemented in a mathematical model and was proven to work [134].

It should be noted that although chloride homeostasis in cultured astrocytes has been well studied [135,136], corresponding in situ experiments are rare and the results are controversial. In this light, a recent study [137] reports that the intracellular concentration of chlorides in astrocytes during sleep is significantly higher and more stable than during wakefulness, contrary to the mechanism described above.

From the point of view of cleaning the extracellular space from harmful metabolites, swelling of the astrocyte during wakefulness and a decrease in its volume during sleep are critical properties. Therefore, the actual role of chlorine and its dynamics are important and require further clarification.

It was multiply reported that aquaporin 4 (AQP4) water channels on the endfeet of astrocytes optimize brain fluid movement and waste clearance [18,138,139]. The deletion of AQP4 channels in transgenic mice eliminates the brain fluid transport and the clearance of proteins from the brain [139,140]. However, the specific role of AQP4 in the regulation of the BWRS functions is still under discussion [141,142,143,144,145].

The new findings in the field suggest a change in the views on the role of AQP4 aquaporins. From the increased expression on astrocytes, the endfeet were interpreted as a probable pathway for the passage of water through the parenchyma.

However, recent results indicate that AQP4 is not required for increased astrocyte volume in either condition: neither hypoosmolar interstitial fluid [146] nor an increased extracellular concentration of potassium [106]. At the same time, AQP4 seems to be important for getting rid of excess water at, say, astrocyte shrinking during the transition from the high to low activity state.

In most mammalian cells, plasma membranes are water permeable, as water can diffuse across the lipid bilayer [147,148,149]. In [111], it is hypothesized that this water entry mechanism (simple diffusion across the cell membrane) may contribute significantly to astrocyte shrinking or swelling. This passive water movement can be fast, as significant volume increases occur in neurons and astrocytes within 1 min in hypo-osmolar conditions using standard solution perfusion rates (1–1.5 mL/min) [150].

Note that AQP4 expression depends on the phase of the circadian cycle [28]. How important this is for the regulation of astrocyte volume during the transition between sleep and wakefulness is currently unknown.

Thus, although all the details are far from being clear, it can be considered an established fact that switching NGVU to a state of low activity is accompanied by a significant change in the intercellular volume, which is critical for BWRS activation. Below we will discuss what is known about the signal that controls such a switch.

## 5. Noradrenaline: Master Regulator or Trigger?

At the systemic (whole-brain) level, the transition between sleep and wakefulness is due to the switching of activity between different groups of neuronal nuclei. When we wake up, the ventrolateral preoptic nucleus decreases its activity, while the monoamine nuclei increase their activity [91,92].

An important part of the structure active during wakefulness is the locus coeruleus (LC) nucleus, which has numerous projections throughout the brain [151,152], where it delivers noradrenaline (NA) [153]. LC release is achieved largely via non-junctional varicosities [154]. Specifically, NA released from small spherical enlargements (about 1 μm in diameter) and spaced at short intervals (1–3/μm) [155].

In the sleep state, LC activity is low, and astrocytic calcium responses are blunted or absent [156,157]. When awake, noradrenaline levels are significantly elevated compared to sleep and vary significantly with activity levels [158]. The work [159] found a correlation between LC activity and increased functional connectivity in human subjects using pupillometry. Pronounced calcium reactions of astrocytes to NA were observed; e.g., during locomotion [160].

Even larger changes in NA levels accompany the transition from sleep to wakefulness [85,161]. In [85], the authors describe the complexity of astrocyte responses to NA, including changes in cell volume in response to activation of noradrenaline receptors. The involvement of astrocytes in the formation of responses to NA both in the stark shift from sleep to awake and perhaps also in other behavioral states is marked in [162].

An intriguing question, however, is that the gradual change in NA levels during wakefulness responds to a smooth change in activity and does not seem to be related to the intensity of drainage. At the same time, a drop in NA below a certain level puts the system into a state of consistently low activity and, obviously, less sensitivity to neuronal excitatory signals. In life systems, these kinds of threshold reactions are typically organized as biological triggers, with positive feedback (or two reciprocal inhibitory couplings, which are the same). At the system level, such a mechanism is known—this is the above-mentioned switching between ventrolateral preoptic and monoamine cores [91,92]. Are there signs of a similar mechanism at the cellular level, within, for example, the NGVU population as well?

In [163], the GANE (“glutamate amplifies noradrenergic effects”) mechanism was proposed, according to which adrenergic hot spots, activation centers, are formed. Although this hypothesis is designed to explain the attention span at the behavioral level, the authors proposed specific cellular mechanisms.

With some simplifications, the GANE pathways are illustrated in Figure 2, where the involved couplings are numbered and explained in the caption. An analysis of these paths reveals not one, but three feedback loops that form a local NA-switch.

(i) Firstly, glutamate spillover reaches the closely located NA varicosite and amplifies NA release (2), which in turn stimulates the presynaptic terminal (3). (ii) Further, released NA reaches the astrocyte (4) and cooperates with glutamate in IP3 production, thus triggering astrocytic calcium response. The activated astrocyte releases gliotransmitters, including D-serine, which is a co-agonist of NMDA receptors at the NA varicosite (5). This is the second positive feedback loop, evidently more slowly activated and longer lasting. (iii) Finally, autoreceptors at NA varicosities make the transition between low and high NA levels more sharp.

Taken together, these mechanisms are able to provide a sharp switch between low and normal activity states—the local sleep switch that is necessary for NGVU to also become a sleep unit. There are still enough questions as to what extent the proposed hypothesis is reliable at the level of an individual NGVU. However, direct modeling of the proposed set of regulation pathways shows that the GANE hypothesis is workable [164]. This, in turn, means that the response to noradrenaline signaling of two small populations or even two nearby NGVUs can be different. All of this offers a logical basis for understanding the key role of the astrocyte in the mechanism of local sleep at the level of the individual NGVU.

Bar et al. investigated the global effects of the neuromodulator norepinephrine (NE) on neuron-astrocyte network communication in co-cultures of neurons and astrocytes and in isolated astrocyte networks [165]. The combination of electrical stimulation and noradrenaline application was used to activate the glutamate-mediated pathway in the elementary neuron–astrocyte networks, each being an experimental model of NGVU. It was found that noradrenaline action causes a marked rise in calcium signaling in astrocytes, while neuronal spontaneous activity is reduced, and the synchrony between elementary neuron–astrocyte networks is perturbed. In general, the results described in [165] at least do not contradict the GANE hypothesis. In particular, they confirm the spatial heterogeneity of the response to NA, even with its global application. Although they did not measure the cell volume directly, the morphological changes found in astrocytes upon application of NA can be interpreted as a swelling of thin astrocyte processes in the presence of NA.

Recently, the authors of [166] concluded that astrocytic NE signaling acts as a separate neuromodulatory pathway that regulates the cortical state and links excitation-related desynchrony with cortical circuit resynchronization.

## 6. Physical Implications for Metabolite Clearance and Prospects for Future Research

While an accelerated removal of substances from parenchyma during sleep has been demonstrated experimentally [11], there is still debate about which specific physical mechanism is responsible for this.

In particular, advection is considered, that is, the transfer of molecules of a substance with a fluid flow, in contrast to the classical diffusion of molecules in a porous medium, which is the parenchyma.

For a detailed analysis of the essence of the issue and the resulting contradictions, we refer to recent reviews on the topic [141,167,168], where arguments for and against both mechanisms are provided. The authors of this review are convinced that only a direct visualization of flows in the parenchyma, which is not yet available, would resolve this dispute. However, regardless of which mechanism is dominant, it is possible to assess how significant the above-discussed changes in the extracellular volume of the neuro–glia–vascular unit are in terms of the relative contributions of advection and diffusion.

On the one hand, diffusion in brain tissues has been purposefully studied in recent decades [48,169,170,171,172,173]. Assuming the classical nature of the process, it is conventional to quantify the properties of parenchyma tissues with the effective diffusion coefficient, which takes into account the tortuosity of the channels and other features of the intercellular space. Recently, [174] drew attention to the fact that diffusion in the brain parenchyma has signs of a more complex process. The so-called Brownian-non-Gaussian diffusion has recently been studied in theoretical physics [175]. Generally, the diffusion flux is determined by the concentration drop and the area of some average cross-section of the channels.

On the other hand, the intensity of the transport of substances by advection is based on the Poiseuille–Hagen law and its varieties [176,177]. Unlike diffusion, the intensity of this process essentially depends on the shape of the cross-section of the channels. In addition to the canonical cases of a round pipe and Couette flow (flow in a flat slot) [178], many variants and complicating factors are possible, which were investigated on specific applications [179,180,181,182].

On the topic of our review, using basic algebra, we can make rough estimates of how a twofold (from 10% to 20%) increase in the ECS share can change the diffusion and advection flows. Our reasoning is illustrated in Figure 3.

Consider a tube of length *L* with radius *R* and cross-section $\pi R^{2}$. The volume of this tube will be $V=L\pi R^{2}$. Its increase by two times corresponds to a twofold increase in *S*. If, in this case, the concentration difference does not change, then the diffusion flux will also increase by a factor of two. If, however, the number of molecules is taken constant, then an increase in volume will lead to a decrease in concentration and the diffusion flux will not change (see panel (a) in Figure 3).

According to Poiseuille’s law, the advection flow in this case will increase by four times, in proportion to $R^{4}$. In this case, the hydrostatic gradient is assumed to be constant. A similar calculation for flat slots instead of round channels will give an increase in flows by eight times, respectively (see panel (b) in Figure 3).

A more realistic account of the 3D ECS structure will make this ratio less impressive, but will not cancel the main result: an increase in the ECS during sleep can significantly change the relative contribution of the mechanisms in favor of advection. Thus, the change in astrocyte volume is indeed a key factor that determines the effectiveness of the removal of harmful metabolites at the local level.

The discussion above suggests an obvious problem for future research. Panel (c) in Figure 3 shows that the model approximation of the real structure of the extracellular space can be different. Thus, it would be of great interest to experimentally quantify simultaneous changes in hydrodynamic characteristics and diffusion properties coupled with a change in the amount of water in tissues. Both data on structure-controlled tissue phantoms [183] and in vivo measurements using recently developed methods [140,184] will be of great interest.

## 7. Problems and Issues in Research of Gender Differences of Night Life of Astocytes

An overwhelming number of studies on mechanisms of sleep have been conducted on male rodents. This is due to the fact that sex hormones in females, depending on the cycle, significantly affect the work of internal organs and the brain. However, there are also sex differences in the development of AD and sleep disorders. One can expect that sex differences may also contribute to the mechanisms of sleep regulation and BWRS function.

In particular, sex differences are significant when it comes to the risk of developing AD. For example, in [185], Zhu et al. list a number of reasons that lead to different frequencies of AD in men and women, including brain parameters, sleep disorders, hormonal differences, vascular disorders, etc. The urgent need to understand how sexual dimorphism should be taken into account when diagnosing AD is also under discussion in the field of psychiatry [186]. There are significant differences in sleep disturbances and quality between men and women [8,21,22,25].

At the same time, the possible contribution of sexual dimorphism becomes much less obvious if we move to the local level of BWRS operation, where such parameters as the geometry of the perivascular space are important, and in relation to the topic of our review—the signaling pathways of astrocytes—the nature of their response to changes in neuronal activity, etc.

One of the rare focused studies on this issue was explored by Gianetto et al. [187]. The main result they obtained is expressed in the paper “Biological sex does not predict glymphatic influx in healthy young, middle aged or old mice”. This conclusion was made based on observations of the distribution of fluorescent markers.

At the same time, very recent results from a human study by Han et al. revealed significant sex-specific changes of the gBOLD-CSF coupling, as a measure of glymphatic function, over a wide age range [188]. Despite the fact that these measurements have been made on large areas of the brain, that is, in fact, at the system level, they raise the question again: Are there differences in astrocyte function between male and female laboratory animals?

The reality is that the bulk of the experimental studies we cited, obtained using laboratory animals, do not allow us to give an unambiguous answer since they were carried out without taking into account the sex of the animals. A noticeably smaller part of the work is based on the use of only males [11,17,35,60,62,84,101,105,128,154,155,156,157].

Few studies take into account the possible dependence of the results on gender but report that no significant differences were found [139,140].

Human studies typically included mixed groups of subjects. As an exception, we note the study [5] effect of one night of total sleep deprivation on beta-amyloid accumulation, and the work [12] where the dependence of slow global CSF oscillations on gender was tested and no differences were found.

Thus, it seems reasonable to assume that sexual dimorphism is clearly significant for system-level processes (sleep, risk of developing AD) but has little, if any, effect on the functioning of small cellular structures, such as neurogliavascular units. However, this fact cannot be considered proven since there is not enough targeted research.

## 8. Conclusions

In this review, we aimed to highlight a chain of facts and mechanisms linking together the transition between sleep and wakefulness, on the one hand, and changes at the level of the smallest functional unit of the brain parenchyma—the neuro-glia-vascular unit. In our opinion, this connection becomes more understandable if we take into account the recently emerged concept of local sleep, which actually suggests considering sleep as a spatially inhomogeneous process. Together with the new understanding of the switching mechanism of action of noradrenaline, a coherent, albeit still incomplete, picture of the relationship is emerging, and awaiting further research.

## Figures and Tables

**Figure 1 cells-12-02667-f001:**
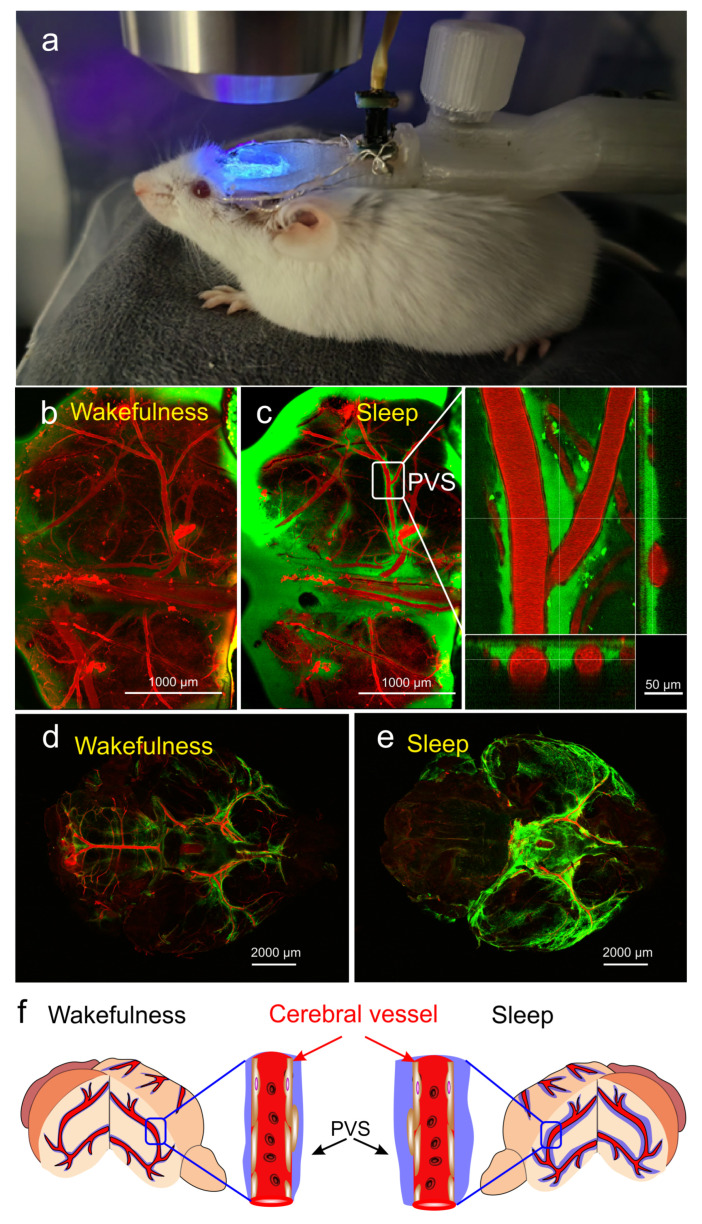
The changes in activity of the brain waste removal system (BWRS) during sleep and wakefulness: (**a**) photo of real-time multiphoton monitoring of BWRS in non-anesthetized mouse under EEG control. (**b**,**c**) Representative images of real-time multiphoton microscopy of fluorescein isothiocyanate-dextran (FITCD, green) distribution in perivascular spaces (PVSs) surrounding the cerebral vessels filled with Evans Blue dye (EBD, red) after its injection into the right lateral ventricle in awake (**b**) and sleeping (**c**) male mouse under EEG control. During wakefulness, PVSs are not filled with FITCD and appear empty. However, during sleep, PVSs are completely filled with FITCD. (**d**,**e**) Representative ex vivo confocal images of FITCD distribution in the brain after its injection into the right lateral ventricle in awake (**d**) and sleeping (**e**) mice. The intensity of fluorescent signal from FITCD is higher in sleeping vs. waking brain. (**f**) Schematic illustration of changes in PVS size during wakefulness and sleep.

**Figure 2 cells-12-02667-f002:**
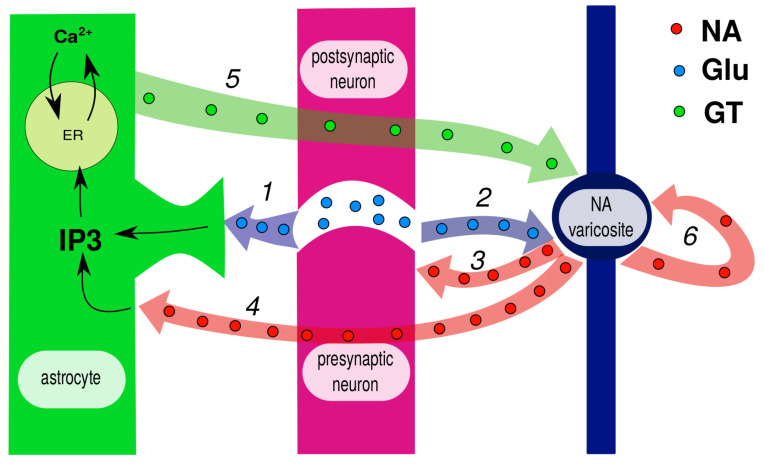
Simplified chart of pathways according to [163]. Glutamate spillover reaches both astrocyte (1) and closely located noradrenaline (NA) varicosite (2). Released NA activates presynaptic β-adrenoreceptors (3) and thus promotes further glutamate release. When astrocyte reaches membrane, NA cooperates with glutamate to activate calcium response via inositol triphosphate (IP3) production (4). Activated astrocyte releases gliotransmitters, including D-serine, which is a co-agonist of N-methyl-D-aspartate (NMDA) receptors at NA varicosite (5). This, in turn, promotes further NE release. Autoreceptors at NA varicosities inhibit its release at low levels but amplify it at high levels and thus serve as neural gain amplifiers (6).

**Figure 3 cells-12-02667-f003:**
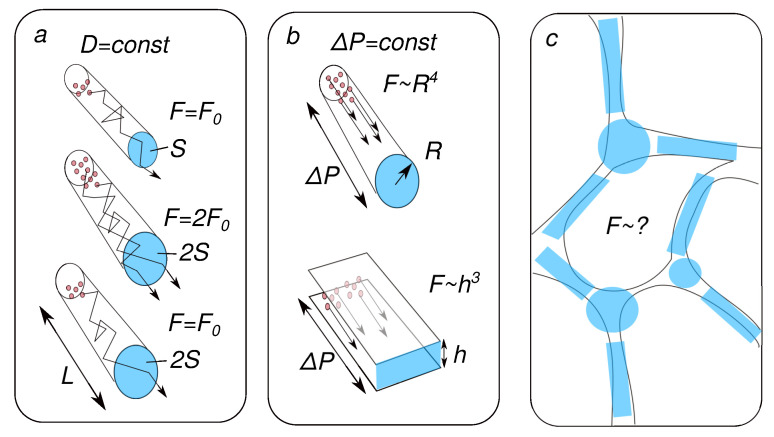
Extracellular space (ECS) fraction and molecular transport, physical background. (**a**) The diffusion flux is proportional to the channel cross-section if the concentration difference is constant. However, if the amount of substance is constant, then increasing the volume of the channel will not increase the flow. (**b**) The flow during advection (transport of particles by liquid) depends on the shape of the cross-section of the channels. (**c**) The specific choice of the approximating method is determined.

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
