# Peer review of "Mechanisms of Activation of Brain’s Drainage during Sleep: The Nightlife of Astrocytes"

_cells, 2023, doi:10.3390/cells12222667_

Round 1

Reviewer 1 Report

Comments and Suggestions for Authors

General Comments:

The aim of this review article is to summarize current knowledge of the role of astrocytes in the activation of the brain waste removal system (BWRS) during sleep, while also proposing mechanisms for the relevant processes and exposing gaps in the research that still need to be investigated. During states of wakefulness, astrocyte volume is larger, shrinking perivascular spaces. During sleep, astrocyte volume shrinks, and a larger volume of interstitial fluid can flow through the now larger perivascular spaces, removing accumulated waste. Evidence for this astrocytic modulation of sleep is presented, followed by the proposal of the neuro-glia-vascular unit (NGVU) as a local unit of sleep modulation and thus also a local unit of the BWRS. Astrocytes are proposed to control the local sleep switch of the NGVU via communication with larger sleep structures in the brain followed by subsequent morphological changes, for which mechanisms are suggested. Finally, an argument is presented for a greater role for advection in metabolite clearance than diffusion based on an astrocytocentric model of the BWRS. 

A strong argument for an astrocytocentric model of the BWRS is presented, along with proposed mechanisms for the relevant phenomena in a logical manner, other than the role of astroglial chloride, for which the research gap is slightly unclear (see minor modifications below). 

Of particular strength is the argument for the NGVU as a local sleep and drainage unit. This was followed by the resourceful suggestion that the “glutamate amplifies noradrenergic affects” (GANE) mechanism, originally proposed to explain behavioural attention spans, may act as a local sleep switch, connected to the systemic sleep switch (LC nucleus) through noradrenaline varicosities.

This review gathered compelling evidence for an astrocytocentric BWRS while also uncovering gaps in the research such as which astrocytic signalling pathways are important in the morphological changes needed for the NGVU to function as a drainage unit.

Significance: Understanding the mechanisms behind the BWRS may be relevant for therapeutically treating AD, for which there is still no cure. However, it may be even more applicable regarding preventative measures, which might be worthwhile to acknowledge in the introduction. 

I support publication of the manuscript after the following changes. 

Minor modifications:

-        Line 18 – All examples of organisms given are animals with a nervous system. Although all organisms may display a form of “sleep”, bacterial periods of dormancy are not relevant to the paper, and so the opening line should be reworked to remain on topic. A working definition of “sleep” should also be provided in the context of the paper. 

-        Section 2.2. Astroglial chloride begins by discussing how chloride plays a role in astrocyte morphological plasticity, suggesting that uptake of K+ during periods of high synaptic activity is accompanied by influx of Cl- and water, resulting in astrocytic swelling. However, the only in situ experimental evidence in the context of sleep is presented at the end of this section and does not agree with the mechanism discussed at the beginning. While this mechanism may be relevant in some physiological situations, as it is stated to be proven by a mathematical model, is not an experimentally proved mechanism of modulation of sleep by astrocytes. Therefore, it should be removed from Section 2 and is better suited for Section 4. Astrocyte volume regulation, where it is also discussed. 

-        Figure 1 – clarify on the labels of Figure 1g, 1j, and 1m whether the AD brain is in a state of wakefulness or sleep, as well as in the figure caption. 

-        Line 108-109 – Astrocytic calcium can be considered an indicator of the sleep/wakefulness state of the brain, not sure about considering the astrocyte to be “asleep” unless a working definition of sleep is given that allows the astrocyte to be classified as “asleep”.

-        Line 169-187 – These paragraphs are within Section 2.3 Adenosine-mediated pathway, but they discuss mechanisms of astrocytic sleep modulation other than adenosine signalling, including K+ signalling, GPCR signalling, and Ca2+ signalling. These should be given their own section or redistributed within the other Section 2 subsections. 

-        Line 185-187 – It is unclear why it can be assumed that the ability of the astrocyte to communicate with the brain structures that control sleep is proportional to neuronal activity and why its purpose is to report local depletion of metabolic resources. Because of gliotransmission? Please elaborate. 

-        Are all of the terms ISF space, PVS, and ECS needed, or can they be consolidated into a single term? Although they can mean different things, in the context of the paper it always seems to be referring to perivascular space between astrocytes and endothelial cells through which interstitial fluid can flow, and it is not clear why the term changes throughout the paper. E.g., Line 43-44: ISF space (interstitial fluid space); Figure 1 caption: PVS (perivascular space); Line 232: ECS (extracellular space). 

-        Line 327 – Would astrocytes hypoosmolar to the solution not decrease in volume? 

-        Line 324-328  If AQP4 is only important for getting rid of excess water, does this mean it could still play a role in astrocyte shrinking during sleep states? 

-        Choose only one of noradrenaline/norepinephrine to use in Section 5.

-        Grammar/syntax to be edited throughout but mostly in introduction. 

Comments on the Quality of English Language

-        Grammar/syntax to be edited throughout— but introduction must be reworked. 

Reviewer 2 Report

Comments and Suggestions for Authors

See attached file.

Comments on the Quality of English Language

Some sentences are unclear. I have tried to highlight as many as I can but would recommend having another person review the manuscript for grammar and sentence structure clarity. 
